# EXOSC10 is required for RPA assembly and controlled DNA end resection at DNA double-strand breaks

Judit Domingo-Prim [1], Martin Endara-Coll[1], Franziska Bonath [2], Sonia Jimeno[3,4], Rosario Prados-Carvajal[3,4], Marc R. Friedländer [2], Pablo Huertas [3,4] & Neus Visa [1]

The exosome is a ribonucleolytic complex that plays important roles in RNA metabolism. Here we show that the exosome is necessary for the repair of DNA double-strand breaks (DSBs) in human cells and that RNA clearance is an essential step in homologous recombination. Transcription of DSB-flanking sequences results in the production of damage-induced long non-coding RNAs (dilncRNAs) that engage in DNA-RNA hybrid formation. Depletion of EXOSC10, an exosome catalytic subunit, leads to increased dilncRNA and DNA-RNA hybrid levels. Moreover, the targeting of the ssDNA-binding protein RPA to sites of DNA damage is impaired whereas DNA end resection is hyper-stimulated in EXOSC10-depleted cells. The DNA end resection deregulation is abolished by transcription inhibitors, and RNase H1 overexpression restores the RPA recruitment defect caused by EXOSC10 depletion, which suggests that RNA clearance of newly synthesized dilncRNAs is required for RPA recruitment, controlled DNA end resection and assembly of the homologous recombination machinery.

[1] Department of Molecular Biosciences, The Wenner-Gren Institute, Stockholm University, SE-106 91 Stockholm, Sweden. [2] Science for Life Laboratory, Department of Molecular Biosciences, The Wenner-Gren Institute, Stockholm University, SE-106 91 Stockholm, Sweden. [3] Centro Andaluz de Biología Molecular y Medicina Regenerativa-CABIMER, Universidad de Sevilla-CSIC-Universidad Pablo de Olavide, 41092 Sevilla, Spain. [4] Departamento de Genética, Universidad de Sevilla, 41080 Sevilla, Spain. Correspondence and requests for materials should be addressed to N.V. (email: neus.visa@su.se)

D NA double-strand breaks (DSBs) are cytotoxic lesions that threaten genomic integrity. The DNA damage-response (DDR) recruits the DNA repair machinery to DSBs and activates checkpoint pathways to stop the progression of the cell cycle until the DNA integrity has been restored[1]. The MRE11-RAD50-NBS1 (MRN) complex recognizes the DSB and recruits the ATM/ATR kinases, which are responsible for the phosphorylation of the H2AX variant histone at Ser137. This phosphorylated histone, γH2AX, acts as a recruitment platform for adaptor proteins and promotes chromatin remodeling to increase the accessibility of the DDR effectors[2,3].

The two major DNA repair mechanisms for DSBs repair are homologous recombination (HR), mostly active in the mid-S and G2 phases of the cell cycle when the sister chromatids are available for faithful repair, and non-homologous end joining (NHEJ), which is considered error-prone but is active through the entire cell cycle[2,3]. The choice between the two pathways is tightly regulated by the initiation of DNA end resection, a 5′–3′ degradation of one strand of the DNA at each side of the break, which is mainly controlled by MRN and CtIP[4]. Once initiated, DNA resection prevents canonical NHEJ and commits the DSB towards the HR pathway. Long-range 5′–3′ DNA end resection is then catalyzed by EXO1 or DNA2, and the resulting single-stranded DNA (ssDNA) tracks become rapidly coated by the ssDNA-binding protein RPA[4,5], which promotes resection termination[6]. RPA is eventually replaced by the strand-exchange factor RAD51, a central player in HR that directs sister-chromatid strand invasion[7].

In parallel with the resection process, RNA polymerase II (RNAPII) transcribes the sequences that surround the DSBs[8–10] and, in *Schizosaccharomyces pombe*, this non-canonical transcription has been shown to regulate DNA resection[8]. Little is known about the interplay between transcription and DNA resection in higher organisms. In mammalian cells, transcription in the vicinity of DSBs leads to the formation of damage-induced, long non-coding RNAs (dilncRNAs) that are processed into short non-coding RNAs[9,10] termed either DNA damage-response RNAs (DDRNAs) or damage-induced RNAs (diRNAs). Recent reports have proposed that diRNAs are required for the assembly of DDR foci and for DNA repair by HR[11–15], but these small RNAs have not been detected at non-repetitive genomic loci and their functional significance is controversial[10,16]. Regardless of whether small diRNAs are universal or restricted to repeated sequences, there is compelling experimental support for the occurrence of de novo transcription by RNAPII at DSBs, both in repetitive sequences and unique genomic sites[8–10,16]. Moreover, different RNA processing factors are recruited to DSBs, which suggests that RNA plays a role in DNA repair. These include the NEXT complex[17], the nuclear poly(A)-binding protein[18], the C1D family proteins[19], helicases such as DDX1[20] and senataxin[21], the decapping protein EDC4[22], and exoribonucleases such as XRN1[23], XRN2[24], and the RNA exosome[23,25,26].

The RNA exosome is a multiprotein ribonucleolytic complex that participates in RNA processing and degradation[27–29]. The exosome has also been linked to RNAPII backtracking and transcription termination[30–32]. This complex is composed of a nine-subunit core that is catalytically inactive and two catalytic subunits, EXOSC10 and DIS3, that can interact with the core independently of each other. EXOSC10 is the ortholog of the yeast and *Drosophila* RRP6, is located predominantly in the cell nucleus and has 3′–5′ exoribonuclease activity. DIS3, also known as RRP44, is both nuclear and cytoplasmic and has exoribonuclease and endoribonuclease activities[27,33].

In *Saccharomyces cerevisiae*, an interaction between RRP6 and the non-canonical poly(A) polymerase TRF4 is required to recruit RPA to ssDNA[23]. In *D. melanogaster*, RRP6 interacts with RAD51 and is required for proper repair by HR[25]. In human cells, an interaction between RAD51 and EXOSC10 was also revealed by a proximity ligation assay[25], but the mechanisms by which the RNA exosome is involved in the HR pathway are poorly understood. Here, we show that the catalytic activity of the exosome subunit EXOSC10 contributes to the HR pathway by degrading dilncRNAs and maintaining RNA homeostasis at DSBs. Our results identify RNA clearance at DSBs as a step in the HR pathway that is required for the assembly of RPA onto the resected ssDNA, which in turn is a prerequisite for controlled DNA resection, RAD51 replacement and DNA repair by HR.

## Results

**The exosome is necessary for DNA repair by HR.** We chose to use laser micro-irradiation to study the recruitment of exosome subunits to sites of DNA damage because the RNA exosome has a widespread distribution throughout the cell nucleus. We micro-irradiated HeLa cells with a 365 nm ultraviolet (UV) laser to produce localised DNA damage and immunostained the irradiated cells with antibodies against EXOSC10 or DIS3 to determine whether these exosome subunits are recruited to the damaged area. DIS3 was clearly relocated to the damaged area, where it co-localised with γH2AX (Fig. 1a). EXOSC10 was highly concentrated in the nucleolus, which impaired the imaging of laser stripes. In order to better reveal the micro-irradiated areas, the cells were treated with Actinomycin D at low concentration (40 nM for 1 h) to inhibit ribosomal RNA synthesis and, in this way, reduce the nucleolar concentration of EXOSC10. In cells treated with this low concentration of Actinomycin D, EXOSC10 relocated to the nucleoplasm and the anti-EXOSC10 antibody revealed prominent EXOSC10 staining in the micro-irradiated stripes (Fig. 1a). These observations show that the exosome catalytic subunits are recruited to sites of DNA damage, in agreement with previous reports[25,26].

In another series of experiments, HeLa cells were treated with the RNAPII inhibitors Actinomycin D (high concentration to inhibit RNAPII, 8 μM for 1 h) or Triptolide (10 μM for 30 min) before micro-irradiation and immunostaining. Transcription inhibition impaired the relocation of EXOSC10 and DIS3 to the irradiated areas (Fig. 1b), which suggests that the recruitment of the exosome to damaged DNA requires ongoing transcription.

Next, we knocked down either EXOSC10 or DIS3 in HeLa cells by RNA interference using short-interfering RNAs (siRNA, Supplementary Fig. 1a, b) and investigated the effects of the depletion on DNA repair. In a first set of experiments, HeLa cells were exposed to ionizing radiation (γ-radiation) and the kinetics of H2AX phosphorylation and dephosphorylation was followed by immunofluorescence at 0, 1, 6, and 24 h after irradiation using an antibody against γH2AX. As shown previously[25], depletion of EXOSC10 inhibited γH2AX recovery (85% of unrepaired damage 24 h after irradiation in EXOSC10-depleted cells), which suggests that EXOSC10 depletion severely impaired DNA repair. In contrast, depletion of DIS3 did not affect the kinetics of γH2AX dephosphorylation (Fig. 1c). A clonogenic assay using different radiation doses (from 0 to 6 Gy) showed that EXOSC10-depleted cells were more sensitive to γ-radiation than control cells, which was in accordance with the severe DNA repair defect reported above. Depletion of DIS3 also increased radiation sensitivity, although not as much EXOSC10 depletion (Fig. 1d).

Flow cytometry analyses of the cell cycle were carried out to establish whether the siRNA treatments had any effect on cell cycle progression that could indirectly influence the repair capacity of the cells. However, we did not detect any cell cycle differences in HeLa cells upon depletion of exosome subunits in the conditions of our experiment (Supplementary Fig. 1c).

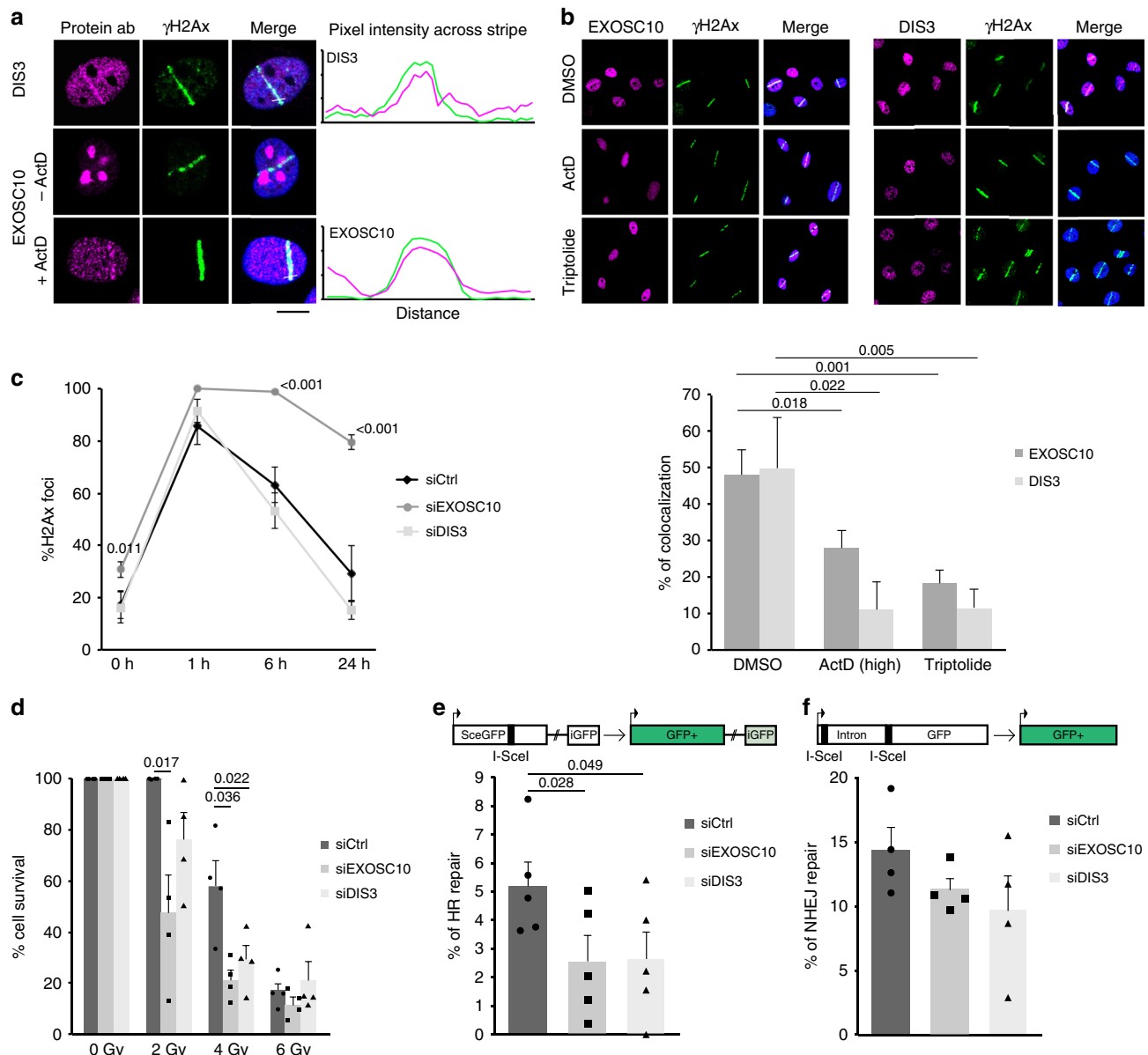

**Fig. 1** The exosome is recruited to DSBs and is necessary for HR. **a** HeLa cells were co-stained with antibodies against either EXOSC10 or DIS3 and γH2AX 30 min after UV laser micro-irradiation. In some experiments, the cells were treated with Actinomycin D at low concentration (40 nM) for 1 h prior to irradiation and immunostaining to inhibit RNA polymerase I. Scale bar: 10 μm. **b** UV laser micro-irradiation experiments were carried out as in **a**, but the cells were treated with either 8 μM Actinomycin D for 1 h or 10 μM Triptolide for 30 min prior to irradiation to inhibit RNAPII. Control cells were treated with DMSO and Actinomycin D (40 nM) to facilitate the imaging of irradiated stripes. The plot shows the percentage of γH2AX-positive stripes co-stained with antibodies against EXOSC10 or DIS3 in each condition. Statistical testing was done using a Mann–Whitney's test and significant p-values are shown. At least 40 cells were analysed in each condition (from three independent siEXOSC10 experiments and two independent siDIS3 experiments). **c** Cells were transfected with either siCtrl, siEXOSC10 or siDIS3, irradiated 48 h after transfection with ionizing radiation (5 Gy), and fixed at different time-points. The percentage of cells that showed γH2AX-positive foci were quantified. The statistical significance was tested as in **b** (n = 3 independent experiments). **d** Clonogenic assay with 0, 2, 4, and 6 Gy after EXOSC10 and DIS3 knock-down. The histogram shows cell survival 7 days after irradiation. p-values were calculated using a paired student t-test (n = 4). **e**, **f** U2OS-DR-GFP or U2OS-EJ5-GFP cells were transfected with either siCtrl, siEXOSC10 or siDIS3 and with a plasmid for I-SceI expression. GFP expression was quantified by flow cytometry and the relative repair efficiencies were calculated. The histograms show the average percentage of GFP-positive cells (% of HR repair in **e** and % of NHEJ repair in **f**) (n = 5 and n = 4 independent experiments in **e** and **f**, respectively). p-values were calculated using a two-tailed Student's t-test. Error bars show s.e.m. Source data for Figs. 1b–f are provided as a Source Data file

In a previous study, we showed that EXOSC10 is required for the recruitment of RAD51 to DSBs[25] and is therefore involved in the HR pathway. To determine whether the exosome was also involved in NHEJ, we used two engineered U2OS cell lines that carry GFP reporter systems for either HR (U2OS-DR-GFP cells)[34] or NHEJ (U2OS-EJ5-GFP cells)[35]. The GFP reporters harbor an I-SceI cleavage site that is used to introduce a sequence-specific DSB. Repair of the DSB restores a functional *GFP* gene the expression of which can be quantified to estimate the DNA repair efficiency (Fig. 1e, f). We depleted either EXOSC10 or DIS3 in the U2OS reporter lines and quantified the effects of the depletions on each of the DNA repair pathways.

To this end, the cells were transfected with a mixture of the I-SceI endonuclease expression plasmid and the siRNAs for either EXOSC10 or DIS3. The percentage of repair in the knock-down was compared to that of control cells transfected in parallel with the I-SceI plasmid and control siRNA. Cells depleted of either EXOSC10 or DIS3 showed a significant reduction of 66% and 40%, respectively, in the HR pathway (Fig. 1e). Instead, no significant differences were observed for NHEJ (Fig. 1f). We also carried out cell cycle analyses by flow cytometry to detect possible effects of the siRNA treatments on cell cycle progression in U2OS cells, which could affect the choice of DNA repair pathway. After DIS3 depletion, and differently to HeLa cells, the cells showed a significantly increased G1 fraction, whereas the S and G2 fractions were reduced (Supplementary Fig. 1d), which could contribute to the lower HR activity observed in DIS3-depleted cells. The EXOSC10-depleted cells did not show any cell cycle alterations.

We concluded that both EXOSC10 and DIS3 are recruited to DSBs and that EXOSC10 is the exosome subunit that is necessary for DSB repair by HR (see Discussion).

**EXOSC10 is required for the recruitment of RPA to DSBs.** We had previously shown that EXOSC10 depletion impairs RAD51 recruitment to DSBs[25]. We carried out micro-irradiation and immunostaining experiments to analyse the recruitment of other HR factors that act upstream of RAD51 and determine whether the failure in RAD51 recruitment was the primary consequence of EXOSC10 depletion or the result of upstream alterations in the HR pathway. HeLa cells were depleted of either EXOSC10 or DIS3, micro-irradiated and immunostained with antibodies against RAD51, RPA or CtIP. An anti-γH2AX antibody was used

to identify the irradiated areas, and the percentage of γH2AX-positive stripes that were co-stained by the antibodies of interest was quantified. As expected, depletion of EXOSC10 significantly diminished the association of RAD51 with the irradiated areas (from 35.6 to 12.6%), whereas DIS3 depletion caused only a slight reduction (from 35.6 to 27.12%) that was not statistically significant (Fig. 2a). The percentage of RPA-positive stripes was also reduced in EXOSC10-depleted cells compared to control cells (from 75.4 to 47.2%) and only slightly decreased (from 75.4 to 67.24%) in DIS3-depleted cells (Fig. 2b). Depletion of EXOSC10 also inhibited the assembly of RPA foci in cells exposed to ionizing radiation (Supplementary Fig. 2). Instead, the percentage of CtIP-positive stripes was not affected by the depletion of exosome subunits (Fig. 2c), which suggests that neither EXOSC10 nor DIS3 are required for CtIP recruitment to DSBs. We concluded that the exosome, or at least EXOSC10, is necessary for a step that is upstream of RPA recruitment but after recruitment of CtIP to the DSB.

**EXOSC10 is required for controlled DNA end resection.** CtIP cooperates with the MRN complex to initiate DNA end resection, and we hypothesized that the observed defects in RPA assembly in EXOSC10-depleted cells could be due to defective DNA end resection. In order to test this possibility, we first performed micro-irradiation experiments. HeLa cells were grown in the presence of bromodeoxyuridine (BrdU) for 24 h, micro-irradiated, and stained using an anti-BrdU antibody without previous DNA denaturation. This method reveals DNA end resection because the BrdU epitope is only accessible in ssDNA. Surprisingly, depletion of EXOSC10 increased the number of cells with BrdU-positive stripes compared to control cells (Fig. 3a), which

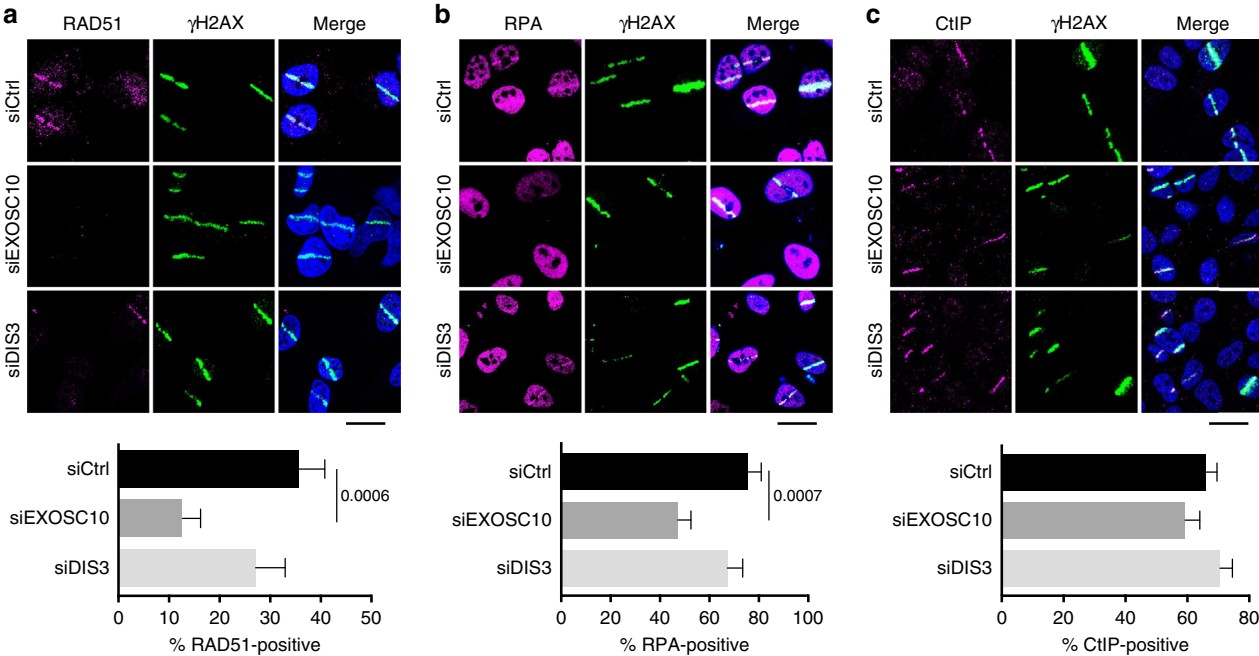

**Fig. 2** Depletion of EXOSC10 impairs RPA and RAD51 recruitment to DSBs. **a** The panel shows RAD51 immunofluorescent staining of HeLa cells depleted of either EXOSC10 or DIS3 for 48 h. The cells were fixed 30 min after UV laser micro-irradiation. The bar plot in the lower part of the figure shows the percentage of γH2AX-positive stripes that were co-stained by RAD51 ($n > 35$ cells analysed in each condition, from two independent experiments). **b** RPA immunofluorescent staining was performed on cells depleted of either EXOSC10 or DIS3 and fixed 15 min after UV micro-irradiation. The bar plot shows the percentage of γH2AX-positive stripes that were co-stained by RPA ($n > 60$ cells analysed for each siRNA treatment, from at least two independent experiments). **c** CtIP immunofluorescent staining was performed on cells depleted of either EXOSC10 or DIS3 and fixed 10 min after UV micro-irradiation. The bar plot shows the percentage of γH2AX-positive stripes that were co-stained by CtIP ($n > 100$ cells analysed for each siRNA treatment, from two independent experiments). In all panels, the error bars represent s.e.m. Statistical testing was done using a Mann–Whitney's test and significant $p$-values are shown in the figure. The scale bars represent 20 μm. Source data for Fig. 2a–c are provided as a Source Data file

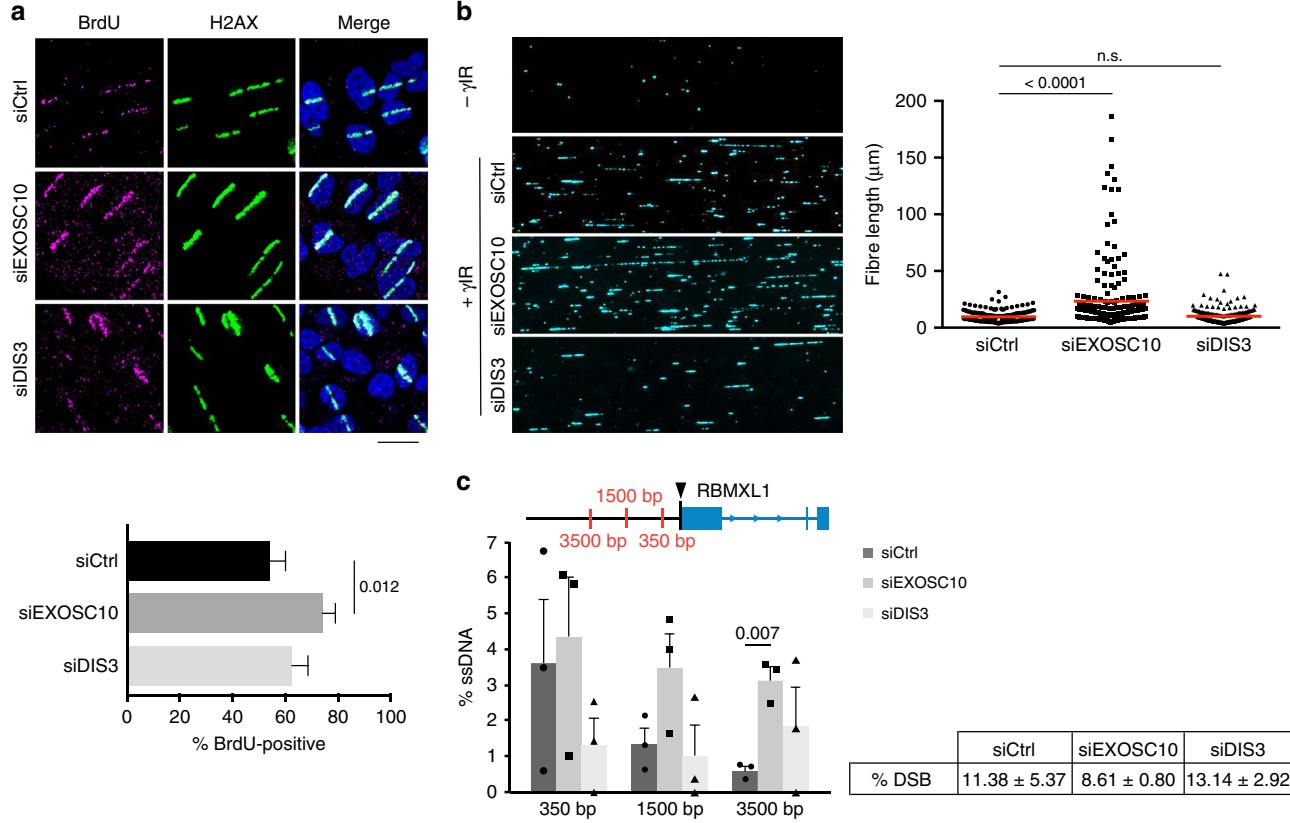

**Fig. 3** EXOSC10 is necessary for limited DNA end resection. **a** HeLa cells were depleted of either EXOSC10 or DIS3, cultivated in the presence of BrdU for 24 h, UV micro-irradiated and fixed for immunofluorescence 10 min after irradiation. The bar plot shows the percentage of γH2AX-positive stripes that were co-stained by an anti-BrdU antibody. Error bars represent s.e.m. (*n* > 60 cells for each siRNA treatment, from at least two independent experiments). Statistical testing was done using a Mann–Whitney's test and significant *p*-values are shown in the figure. The scale bar represents 20 μm. **b** The image shows single-strand DNA tracks (SMART) analysed in U2OS cells 48 h after transfection with siCtrl, siEXOSC10 or siDIS3. The graph shows fibre length quantifications of one out of three biological replicates with *n* = 200 cells analysed. Statistical testing was done using a Mann–Whitney's test and significant *p*-values are shown in the figure. **c** DIvA cells were transfected with either siCtrl, siEXOSC10 or siDIS3 for 48 h and treated with 300 nM 4-OHT for 4 h before extracting the genomic DNA for analysis of DNA end resection. ssDNA levels were measured by qPCR at three positions upstream of the AsiSI cleavage site (*arrowhead*). The non-DSB background levels were subtracted and the corrected ssDNA values are shown in the histogram. The error bars represent s.e.m. from three independent experiments. Statistical testing was done using a paired Student's *t*-test (*n* = 3) and significant *p*-values are shown in the figure. The percentage of DSBs produced in each condition is shown in the figure. Source data for Fig. 3a–c are provided as a Source Data file

suggests that DNA resection is hyperactivated in the absence of EXOSC10. No significant changes were observed after knocking down DIS3 (Fig. 3a).

Next, we asked whether the observed differences in BrdU labelling were due to differences in the length of the resected DNA and we performed a single molecule analysis of resected tracks (SMART). This method is also based on BrdU incorporation and immuno-fluorescence, but allows the visualization, length measurement, and quantification of the individual ssDNA fibers, which is achieved by stretching the DNA by DNA combing[36]. SMART showed a very significant increase of the length of the ssDNA tracks in cells depleted of EXOSC10 compared to control cells, but no differences in ssDNA length were observed in DIS3-depleted cells (Fig. 3b and Supplementary Fig. 3a). In order to confirm that the long ssDNA tracks revealed by SMART in EXOSC10-depleted cells were bona fide CtIP-dependent tracks related to DNA damage, we performed SMART in both irradiated and non-irradiated conditions, and in double-knockdown CtIP-EXOSC10 cells. Indeed, the long ssDNA tracks observed in EXOSC10-depleted cells were induced by DNA damage and CtIP-dependent (Supplementary Fig. 3b).

We also used the quantitative resection assay developed by Zhou et al.[37] to measure the effects of exosome depletion on DNA end resection by quantitative PCR (qPCR). This assay quantifies

ssDNA at sequence-specific DSBs and is based on the use of DSB-induced via AsiSI (DIvA) cells[21]. In DIvA cells, the AsiSI restriction enzyme is expressed as a fusion with an estrogen receptor ligand-binding domain that is efficiently translocated to the nucleus in the presence of 4-hydroxytamoxifen (4-OHT). We knocked down EXOSC10 or DIS3 in DIvA cells, treated the cells with 300 nM 4-OHT and analysed DNA resection at an AsiSI-induced DSB that has been characterised in previous studies (hg38:Chr1:88992917)[37]. The percentage of DNA resec-tion was quantified at three sites located at different distances from the DSB (Fig. 3c). DNA resection in control cells was prominent 350 nt upstream of the DSB (3.6% ssDNA) and was progressively lower at longer distances, as expected, due to resection termination. Depletion of EXOSC10 did not cause any significant change in the percentage of ssDNA 350 nt away from the DSB, and this observation suggests that DNA end resection was initiated normally. However, the levels of ssDNA at 1500 and 3500 nt away from the DSB were significantly higher in EXOSC10-depleted cells than in control cells, which suggests that depletion of EXOSC10 inhibits resection termination.

**DIlncRNAs are substrates of EXOSC10.** Transcription and DNA–RNA hybrids have been shown to regulate DNA end

resection in yeast[8]. Based on this observation and on the fact that EXOSC10 is a ribonuclease, we hypothesized that the exosome regulates transcript levels in the surroundings of DSBs, which in turn affects DNA resection. To directly study dilncRNAs at DSBs, we analysed three different sequence-specific DSBs: the AsiSI site in DIvA cells described above, and two I-PpoI sites in HeLa cells that are located in the *28S* rDNA locus and in an intron of the *Ryr2* gene, respectively. In all cases, we carried out strand-specific reverse-transcription quantitative PCR experiments (ssRT-qPCR) to quantify RNAs transcribed from DSB-flanking sequences in the upstream and downstream directions (Fig. 4a). In control cells with intact exosome activity, the RNA levels upstream of the DSB were very low in the antisense strand, as expected (Supplementary Fig. 4a). However, after cleavage, the transcript levels were significantly higher in EXOSC10-depleted cells than in control cells, both upstream and downstream of the DSBs, in the three analysed loci (Fig. 4b, c), which suggests that EXOSC10 participates in the turnover of dilncRNAs. No increase was observed in the downstream region of the *28S* locus where the

high abundance of sense rRNAs would preclude the detection of newly synthesized sense rRNA transcripts.

We considered the possibility that the changes in RNA levels reported above could be due to differences in the number of DSBs produced in the different experimental conditions. However, the siRNA treatments per se did not affect the frequency of DSB formation (Fig. 3c and Supplementary Fig. 4b).

In summary, our present observations confirm earlier reports on the synthesis of dilncRNAs at DSB[9,10]. Furthermore, our results reveal that dilncRNAs are targets of EXOSC10.

**EXOSC10 reduces DNA–RNA hybrid levels**. A recent study by Cohen et al.[21] showed the formation of DNA–RNA hybrids in DSB-flanking sequences and we asked whether the dilncRNAs that accumulate in EXOSC10-depleted cells can form DNA–RNA hybrids. We performed DNA–RNA hybrid immunoprecipitation (DRIP-qPCR) in control and EXOSC10-depleted cells, and we observed significantly increased DRIP signals both upstream and

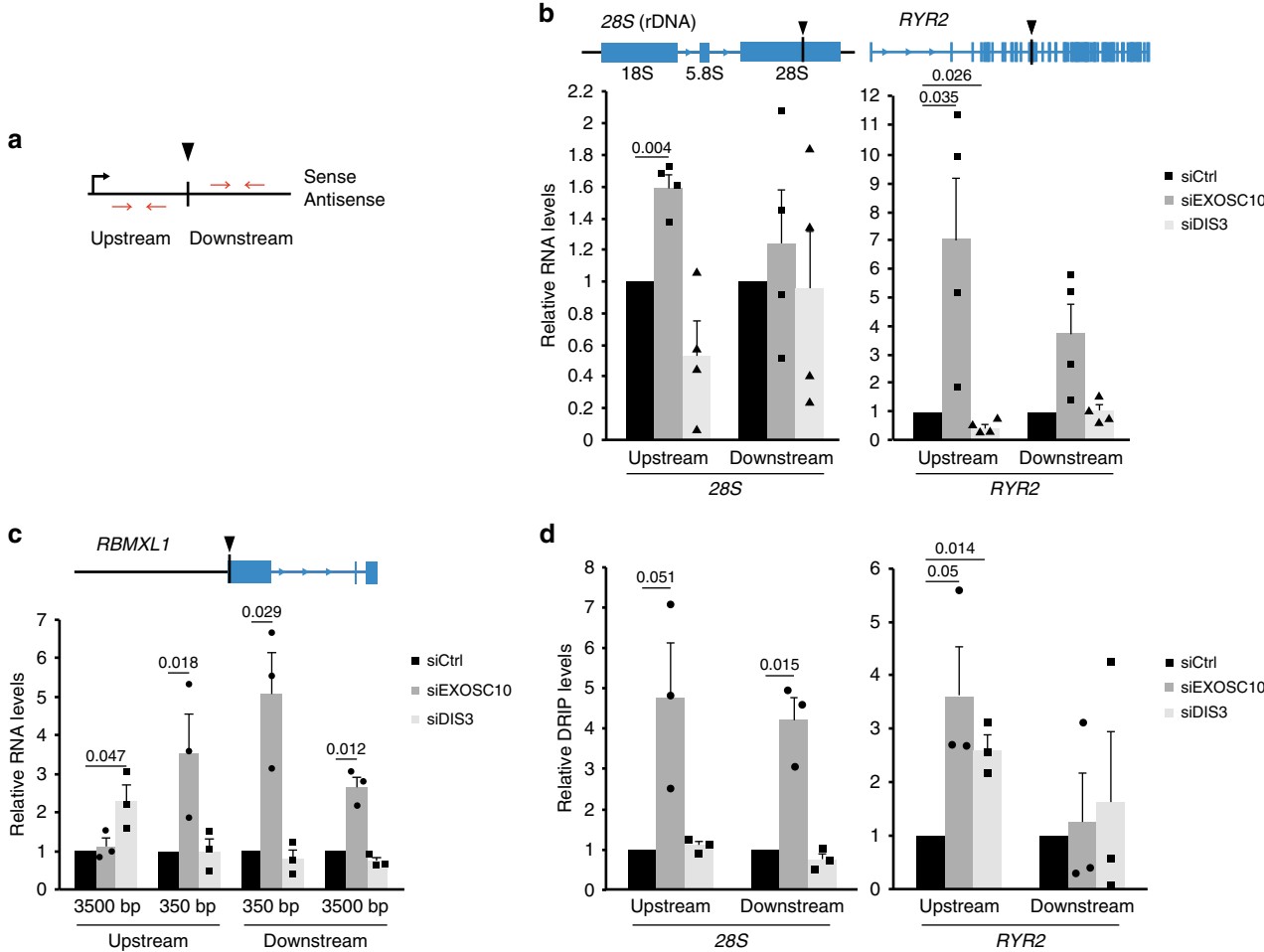

**Fig. 4** DilncRNA levels are increased upon depletion of EXOSC10. **a** Schematic drawing of a DSB showing the nomenclature used to describe the origin of dilncRNAs transcribed from DSB-flanking sequences. **b** SsRT-qPCR experiments were carried out to analyse the levels of upstream antisense and downstream sense (at 500 bp from the DSB) RNAs in the *28S* and *RYR2* genes in HeLa cells, as indicated, 20 h after transfection with the pOPRSVI/MCS-I-PpoI plasmid for I-PpoI expression. The graph shows the ratios cut/uncut for siEXOSC10 and siDIS3 samples, compared to the ratios observed in the control cells (siCtrl). RNA levels were normalized to ARPP. **c** DIvA cells were transfected with siCtrl, siEXOSC10, or siDIS3 and incubated for 4 h with 300 nM 4-OHT before RNA analysis. The graph shows relative RNA levels quantified by ssRT-qPCR as in **b**. **d** DRIP-qPCR was performed in HeLa cells 20 h after transfection with the pOPRSVI/MCS-I-PpoI plasmid. The graph shows the relative DRIP-qPCR levels in cells treated with either siEXOSC10 or siDIS3 compared to control cells (siCtrl). DRIP-qPCR levels were normalized to GADPH. Error bars show s.e.m. from four independent experiments in **b** and **d**, and three in **c**. Statistical testing was done using a one-sample Student's *t*-test and significant *p*-values are shown in the figure. Source data for Fig. 2a–c are provided as a Source Data file

downstream of the analysed DSBs (Fig. 4d). As expected for DNA–RNA hybrids, such signals were sensitive to RNase H1 treatment (Supplementary Fig. 5). These results suggest that, in normal conditions, the exosome degrades dilncRNAs that otherwise engage in DNA–RNA hybrid formation. RNA helicases such as senataxin are recruited to DSBs and can resolve DNA–RNA hybrids at DSBs[21,26]. However, in the absence of EXOSC10, DNA–RNA hybrid resolution does not seem to be sufficient to keep the DNA free from complementary dilncRNA (see Discussion).

Experiments of DIS3 depletion combined with ssRT-qPCR (Fig. 4b, c) and DRIP-qPCR (Fig. 4d) did not support any major role for DIS3 in dilncRNA degradation. Contrarily, DIS3-depleted cells showed significantly reduced dilncRNA levels upstream of the DSB in the *Ryr2* gene, while DNA–RNA levels at the same site were increased (Fig. 4b, c, d). DilncRNA synthesis and R-loop formation has been linked to HR, and DIS3-depleted cells have low HR activity, which could explain the low dilncRNA levels observed in these cells. On the other hand, DIS3 is known to be required for efficient termination of stalled RNAPII complexes, and the effects of DIS3 depletion observed at *Ryr2* could be explained by RNAPII termination defects close to the DSB. To clarify this situation, we carried out RNAPII ChIP-qPCR in cells depleted of DIS3 and observed an increase of RNAPII levels upstream of the *Ryr2* I-PpoI cleavage site (Supplementary Fig. 6). A similar RNAPII increase was observed in cells depleted of TFIIS, a protein required for RNAPII backtracking[31].

These results suggest that DIS3 depletion causes a defect of RNAPII termination upstream of the *Ryr2* DSB, which results in reduced dilncRNA synthesis.

**The exosome controls diRNA levels at DSBs**. We have previously shown that dilncRNAs derived from the *28S* locus upon I-PpoI cleavage are processed into small diRNAs[10]. To investigate whether the exosome affects diRNA levels, we carried out small RNA-seq in HeLa cells transfected with the I-PpoI expression plasmid and we quantified diRNAs in control cells and in cells depleted of either EXOSC10 or DIS3. In non-transfected cells, small RNAs from the *28S* locus were detected in the sense direction, but antisense RNAs were virtually absent in this region, as expected (Fig. 5a, *uncut*). The RNA levels observed in the sense strand were presumably degradation products of the *28S* rRNA. In I-PpoI-transfected cells (Fig. 5a, *cut*), antisense diRNAs complementary to sequences located upstream of the DSB were formed. This very distinct distribution of diRNAs suggested that they are derived from precursor dilncRNAs that start near the DSB and extend in the antisense direction[10]. The absence of antisense diRNAs downstream of the DSB reveals that either antisense transcription in the downstream region does not take place, or that downstream dilncRNAs are not processed into diRNAs.

Increased antisense diRNAs were also detected in EXOSC10-depleted cells where their levels were higher than in control cells.

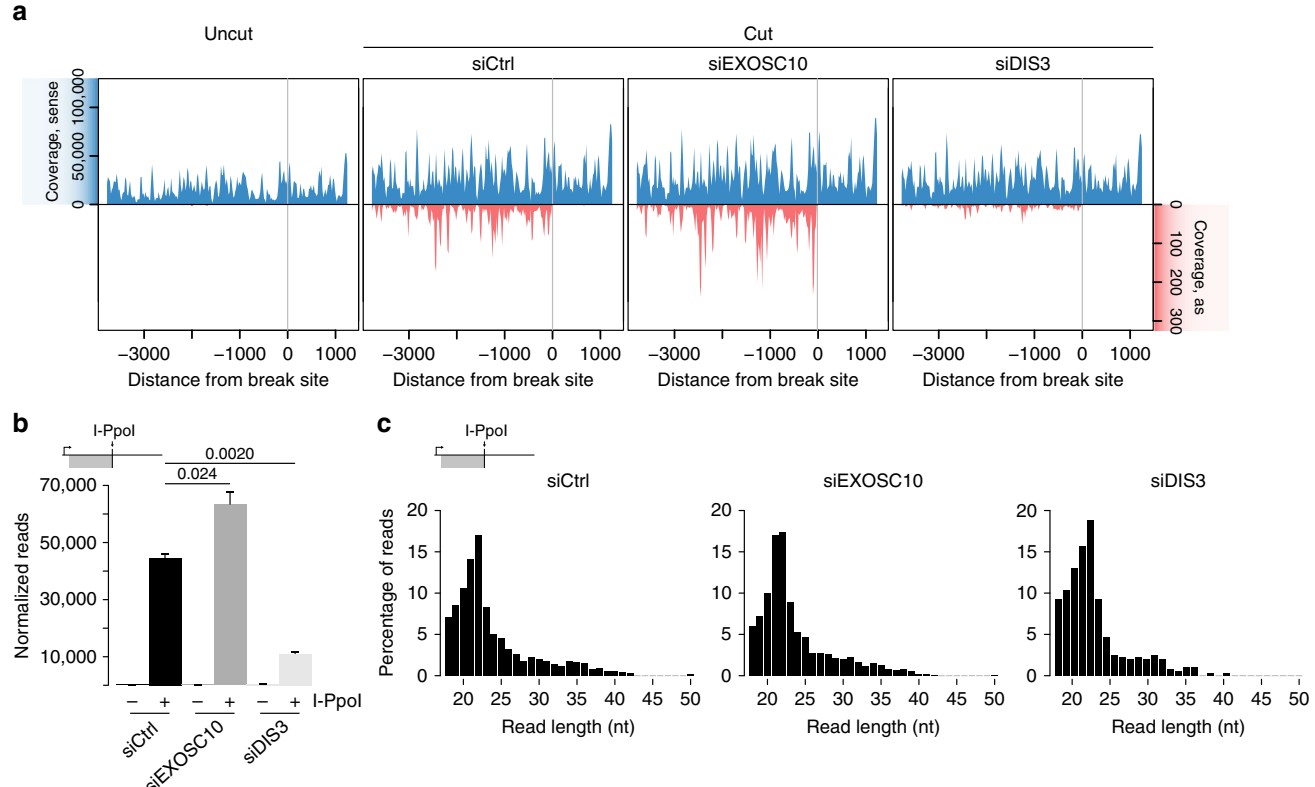

**Fig. 5** Exosome depletion affects diRNA levels but not diRNA processing. **a** HeLa cells were transfected with control siRNA, siEXOSC10, or siDIS3 and the pOPRSVI/MCS-I-PpoI plasmid for I-PpoI expression. Total RNA was purified 36 h after I-PpoI transfection and the small RNA fraction was isolated and sequenced. The plots show the coverage profile of small RNAs that map to the *28S* rRNA locus normalized to spike-in RNAs in non-transfected cells (*uncut*) and transfected cells (*cut*). The x-axis indicates the distance from the I-PpoI cleavage site. **b** The graph shows the number of collapsed reads normalized to spike-ins that appear at the upstream antisense region of the I-PpoI DSB. The error bars show s.d. Changes in read counts in siEXOSC10 and siDIS3 compared to control siRNA are statistically significant using a paired two-tailed Student's *t*-test (*n* = 3). **c** Read length distribution based on the percentage of upstream antisense reads in the three different conditions control siRNA, siEXOSC10 and siDIS3, as indicated. Source data for Fig. 5a–c are provided as a Source Data file

Instead, cells depleted of DIS3 showed much lower diRNA levels (75.4% reduction, Fig. 5a, b). This drastic reduction could be explained by the low HR activity of DIS3-depleted cells and by the reduced dilncRNA synthesis reported above.

The length of the diRNAs was not affected by EXOSC10 or DIS3 depletion, and in both cases the diRNA population showed a prominent peak of 21–22 nt as in control cells that corresponds to Dicer-dependent diRNAs, as reported in a previous study[10] (Fig. 5c).

In summary, EXOSC10 contributes to RNA homeostasis at DSBs by degrading the dilncRNAs that are synthesized in the surroundings of the damaged region. The levels of dilncRNA and diRNA are directly correlated, which suggest that dilncRNA levels are a rate-limiting factor for diRNA production.

**RNA degradation is needed for RPA incorporation to ssDNA.** The results reported above support a model in which EXOSC10 degrades dilncRNAs, and that dilncRNA degradation is required for RPA recruitment, the assembly of the HR machinery and to limit DNA end resection. We constructed a catalytically inactive mutant of EXOSC10[25,38], EXOSC10dea, to directly investigate the role of EXOSC10's catalytic activity at DSBs (Supplementary Fig. 7a). HeLa cells were transfected with plasmids for expression of either wild-type (EXOSC10wt) or mutant EXOSC10 (EXOSC10dea) and analysed after 24 h. The total levels of EXOSC10 were not changed in these conditions, as shown by

western blotting, but the levels of the recombinant mRNAs were 40–60 times higher than those of the endogenous Exosc10 mRNA and two known exosome targets were stabilised in these conditions (Supplementary Fig. 7b–d), which implies that a very large fraction of the total EXOSC10 protein was recombinant. Laser micro-irradiation experiments showed that the amino acid substitutions in EXOSC10dea did not impair the recruitment of the mutant protein to damaged DNA (Supplementary Fig. 8).

In a set of experiments, we expressed the recombinant EXOSC10 proteins in HeLa cells, irradiated the cells and stained them at different time-points with the antibody against γH2AX. The kinetics of H2AX phosphorylation and dephosphorylation in cells that expressed the EXOSC10wt was very similar to that reported for control cells (compare EXOSC10wt in Fig. 6a with siCtrl in Fig. 1c). However, expression of EXOSC10dea caused a significant delay in the recovery of γH2AX levels after ionizing radiation that was visible 6 h after irradiation (Fig. 6a). The levels of γH2AX recovered after 24 h, probably due to the residual activity of endogenous EXOSC10. Moreover, in laser micro-irradiation experiments, the expression of EXOSC10wt rescued the RPA-recruitment defect observed in EXOSC10-depleted cells, but expression of the catalytically inactive EXOSC10dea mutant did not (Fig. 6b). This shows that RNA degradation by EXOSC10 is required for the recruitment of RPA to damaged DNA.

We hypothesized that RNA degradation by EXOSC10 was necessary in order to prevent DNA–RNA hybrid formation and, in

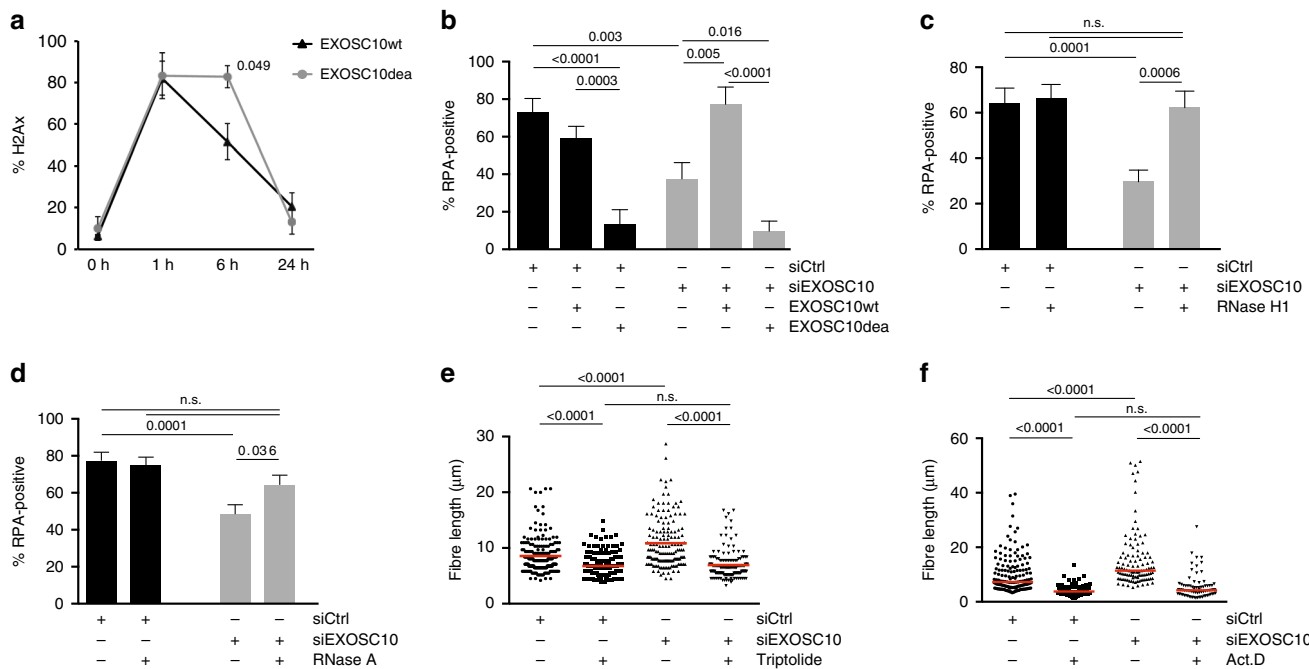

**Fig. 6** DilncRNA degradation is necessary for RPA recruitment. **a** The EXOSC10 catalytically inactive mutant EXOSC10dea or wild-type EXOSC10 were overexpressed in HeLa cells and γH2AX immunofluorescence was performed at 0, 1, 6, and 24 h after exposing the cells to γ-radiation (5 Gy). The graph shows the percentage of cells with γH2AX foci (n = 150 cells analysed at each time point, from three different experiments). **b** HeLa cells were transfected with either siCtrl or siEXOSC10. A second transfection was carried out 24 h later with plasmids for expression of EXOSC10wt or EXOSC10dea, as indicated. The cells were micro-irradiated with a 365 nm UV laser 24 h after the second transfection and the recruitment of RPA to the laser stripes was analysed as in Fig. 2b (n > 20 cells for each condition, from three independent experiments). **c** HeLa cells were transfected with either siCtrl or siEXOSC10. A second transfection was carried out 24 h later with a plasmid for expression of RNase H1 and the recruitment of RPA to laser stripes was analysed (n = 50 cells analysed in each condition, from three independent experiments). **d** HeLa cells transfected with either siCtrl or siEXOSC10 were micro-irradiated with a 365 nm UV laser, permeabilized with CSKT and treated with or without RNAse A for 30 min at 37 °C. The graph shows the percentage of cells with γH2AX stripes that were co-stained with RPA (n > 80 cells analysed in each condition, from three independent experiments). **e**, **f** SMART was done in U2OS cells transfected with either siCtrl or siEXOSC10. The cells were treated with 10 μM Triptolide for 30 min (**e**) or 8 μM ActD for 1 h (**f**) before exposure to γ-radiation (10 Gy). The plot shows the measurements of fibre length from one representative experiment (n = 200 analysed fibres). The averages of the median values from three independent experiments are shown in Supplementary Fig. 9. In all cases, the error bars represent s.e.m. and p-values were calculated with Mann–Whitney's test. Source data for Fig. 6a–f are provided as a Source Data file

this way, facilitate the binding of RPA to the resected ssDNA. To test this possibility, we asked whether overexpression of RNase H1 could restore the RPA assembly defect observed in EXOSC10-depleted cells. We transfected HeLa cells with a plasmid for overexpression of RNase H1 and analysed RPA recruitment in micro-irradiation experiments. As shown in Fig. 6c, RNase H1 overexpression rescued RPA incorporation. Digestion of the cells with RNase A (0,1 mg/ml for 30 min) could also restore the RPA assembly defect observed in EXOSC10-depleted cells (Fig. 6d).

We also carried out SMART experiments in which we inhibited RNAPII transcription with either Actinomycin D (8 μM for 1 h) or Triptolide (10 μM for 30 min) in control cells and EXOSC10-depleted cells prior to irradiation. Both inhibitors abolished the DNA end hyper-resection observed in EXOSC10-depleted cells (Fig. 6e, f and Supplementary Fig. 9), which further supports the conclusion that the resection defect caused by EXOSC10 depletion is due to RNA accumulation.

**Transcription by RNAPII facilitates DNA end resection**. The SMART experiments shown in Fig. 6e, f also suggested that transcription of the DSB-flanking region by RNAPII is required for DNA end resection. We further investigated the link between transcription and resection by measuring resection at a sequence-specific DSB located in an intergenic silent locus in DIvA cells (chr13:105238551; Cohen et al.[21]). The absence of transcripts derived from this locus was confirmed by ssRT-qPCR (Supplementary Fig. 10). Measurements of ssDNA levels in control and 4-OHT-treated cells failed to reveal any significant resection at this locus (Fig. 7a). Moreover, the RNAPII inhibitors Actinomycin D and Triptolide reduced DNA end resection at the transcribed RBMXL1 locus (Fig. 7b).

In another series of experiments, we asked whether transcription was necessary for the formation of RPA foci in HeLa cells exposed to ionizing radiation. The recruitment of RPA to sites of DNA damage was significantly reduced by Triptolide and Actinomycin D (Fig. 7c).

In summary, we concluded that transcription by RNAPII facilitates DNA end resection at DSBs in human cells.

## Discussion

EXOSC10 and DIS3 are subunits of the RNA exosome, a complex that has a variety of roles in RNA metabolism[28]. Here, we reveal that the exosome regulates dilncRNA homeostasis at DSBs. We have shown that the two catalytic subunits of the nuclear RNA exosome are recruited to sites of DNA damage, and a previous study by Richard et al.[26] showed that RRP45, an exosome core subunit, was also targeted to DNA damage sites by interacting with the DNA–RNA helicase senataxin[26]. We conclude, based on these observations, that the entire exosome complex is relocated by the DDR.

We showed in a previous study that EXOSC10 is necessary for efficient HR[25]. Here, we have also studied the contribution of DIS3 to DNA repair. DIS3-depleted cells show increased sensitivity to γ-radiation and the use of a GFP reporter system revealed that DIS3-depleted cells have a decreased HR capacity. However, depletion of DIS3 affects cell cycle progression and delays the entry into S phase, which in itself could explain why DSB repair by HR is reduced in DIS3-depleted cells. On the other hand, our study of γH2AX phosphorylation in cells exposed to γ-radiation did not reveal any decrease in the overall DNA repair capacity, which suggests that DIS3-depleted cells can efficiently repair DSBs by NHEJ.

DilncRNAs are damage-induced RNAs that are synthetized at DSBs by RNAPII[9,10]. DilncRNAs can be processed by Dicer into short diRNAs[10] and it has been proposed that diRNAs are necessary for the activation of the DDR[8,11,14]. However, diRNAs have only been detected at DSBs produced in repetitive genomic regions[10,16], whereas dilncRNAs have been detected at both repetitive and unique sequences[10]. Our study confirms the synthesis of dilncRNAs at three different sequence-specific DSBs, shows that dilncRNAs are targeted by EXOSC10, and sheds light on their functional relevance. Using SMART and analysis of DNA resection at a sequence-specific DSB in human cells, we have shown that transcription is required for the proper regulation of DNA end resection, in accordance with previous reports on the regulation of DNA resection in *S. pombe*[8] and on the preferred choice of repair pathways in mammalian cells[39]. These observations suggest that de novo transcription by RNAPII at DSBs might be needed to facilitate DNA end resection, not necessarily for dilncRNA biogenesis.

Recent RNA-seq and DRIP-seq analyses in DIvA cells suggest that dilncRNAs are synthesized only at transcriptionally active chromatin[21]. In agreement with this proposal, we analysed one DSB produced in an intergenic, non-transcribed genomic sequence and did not find evidence for dilncRNA production at this silent locus.

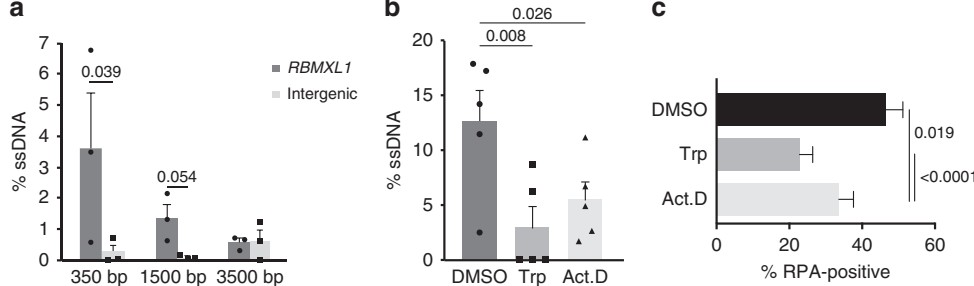

**Fig. 7** Transcription facilitates DNA end resection. **a** DIvA cells were incubated for 4 h with 300 nM 4-OHT and DNA end resection was measured by qPCR at three positions upstream of an AsiSI cleavage site located in a silent, intergenic sequence. The plot compares the percentage of ssDNA at the intergenic locus with that calculated for the transcribed RBMXL1 locus in Fig. 3c. The error bars represent s.e.m. and statistical testing was done using a two-tailed Student's *t*-test (n = 3 independent experiments). **b** DNA end resection at the RBMXL1 locus was also quantified in DIvA cells treated with either 10 μM Triptolide for 30 min or 8 μM ActD for 1 h. The plot shows the percentage of ssDNA 350 nt upstream of the AsiSI cleavage site in control cells (DMSO) and cells treated with RNAPII inibitors. The error bars represent s.e.m. and statistical testing was done using a one-tailed Student's *t*-test (n = 5 independent experiments). **c** The effect of transcription inhibition on the formation of RPA foci was analysed in HeLa cells exposed to ionizing radiation (5 Gy) and stained with antibodies against RPA and γ-H2AX 15 min after irradiation. The bar plot shows the percentage of cells with γH2AX-positive foci that were co-stained by RPA. Error bars show s.e.m. Statistical testing was done using a Mann–Whitney's test (n > 100 cells analysed in each condition, from four independent experiments). Source data for Fig. 7a–c are provided as a Source Data file

The depletion of EXOSC10 leads to increased dilncRNA levels, increased DNA–RNA hybrid levels, reduced RPA targeting to sites of DNA damage, and deregulated DNA end resection. Moreover, the DNA hyper-resection defect is abolished when transcription is inhibited, and RNase A and RNase H1 can restore the RPA-recruitment defect caused by EXOSC10 depletion. These observations suggest a model in which de novo transcription takes place at DSBs to favor DNA end resection, which leads to the production of dilncRNAs in the DSB-flanking sequences. Once dilncRNAs are produced, EXOSC10 is required to degrade them and, in this way, maintain low RNA levels in the vicinity of the DSB to allow RPA incorporation (Fig. 8). Overexpression of catalytically inactive EXOSC10 mutants caused DNA repair defects, which further supports the conclusion that the catalytic activity of EXOSC10 is needed for DNA repair. DilncRNAs can form DNA–RNA hybrids, as shown by us and others[16,21,40], and our results suggest that dilncRNAs can compete with RPA for binding to the resected ssDNA tracks. This proposal is supported by recent in vitro experiments in which RAD52-bound RNA can evict RPA from a complementary ssDNA strand[41].

Two different defects are observed in EXOSC10-depleted cells: inhibition of RPA assembly and extended ssDNA resected tracks. Both defects are probably related to each other. RPA interacts with multiple DNA helicases and translocases and is involved in the initiation of DNA resection by EXO1 and DNA2/BLM[4,42,43]. On the other hand, RPA regulates the activity of EXO1 and promotes resection termination[6,44]. It is reasonable to propose that in EXOSC10-depleted cells, DNA end resection is deregulated and cannot be stopped due to the absence of a well-assembled RPA-ssDNA complex.

Recent transcriptome-wide studies have revealed that EXOSC10 and DIS3 share a large number of substrates[45]. However, the contribution of DIS3 to dilncRNA metabolism is very different from that of EXOSC10. DIS3 depletion results in increased RNAPII levels in the proximity of DSBs and abolishes dilncRNA biogenesis. DIS3 has been involved in transcription termination[31] and we speculate that RNAPII stalling at or near DSBs prevents de novo dilncRNA biogenesis in the absence of DIS3. Further studies will be needed to understand why the depletion of DIS3 results in cell cycle alterations and whether such alterations are linked to defects in DNA repair. However, the fact that DIS3-depleted cells are able to repair DSBs by NHEJ suggests that both dilncRNAs and diRNAs are dispensable for NHEJ.

EXOSC10 acts preferentially on unstructured RNAs in vitro[46] and relies on specialized cofactors to unwind structured substrates in vivo[47–49]. The human exosome cooperates with the helicases senataxin and Mtr4 for the resolution of DNA–RNA hybrids[50,51]. Senataxin is required to resolve DNA–RNA hybrids in DSB-flanking regions, and DNA–RNA hybrid stabilization results in an increased frequency of translocations that are caused by illegitimate ligation between distant DSBs[21]. We have observed a similar stabilization of DNA–RNA hybrids in EXOSC10-depleted cells and our results suggest that both helicase and ribonuclease activities are needed at DSBs to restrain DNA–RNA hybrid formation.

The inhibitory effect of dilncRNAs and DNA–RNA hybrids on HR that we report here is particularly interesting in view of a recent study in which DNA–RNA hybrids are shown to promote HR by contributing to the recruitment of BRCA1, BRCA2 and RAD51 to DSBs[40]. The integrated view that emerges from this study and our present results is that the assembly of the HR machinery is governed by a balance between RNA synthesis and degradation at DSBs: transcription and DNA–RNA hybrid formation are required for the recruitment of DNA repair factors, but DNA–RNA hybrid levels have to be modulated to allow RPA binding and avoid DNA hyper-resection.

An important conclusion from our study is that dilncRNA clearance by the exosome is crucial for HR and for the maintenance of genomic stability. Studies of DNA repair in *Drosophila melanogaster* revealed that the EXOSC10 ortholog, RRP6, is needed for the recruitment of RAD51 to DSBs, and in budding yeast, RRP6 promotes the assembly of RPA-ssDNA complexes[23]. These observations suggest that the function of the exosome in DNA repair is evolutionarily conserved.

## Methods

**Cell culture**. HeLa (ECACC, 93021013, Sigma-Aldrich, U2OS and DIvA cells were cultured with Dulbecco's Modified Eagle's Medium (Gibco, Thermo Scientific) supplemented with 10% fetal bovine serum and 1% penicillin/streptomycin at 37 °C in a humidified incubator with 5% $CO_2$. DIvA cells were maintained in culture medium containing 1 μg/ml puromycin. In some cases, the cells were cultured in the presence of Actinomycin D for 1 h or Triptolide for 30 min, as indicated.

**Short-interfering RNA (siRNA)**. siRNAs were purchased from Invitrogen (Ambion, Custom Select siRNAs). Transfection in mammalian cells was performed with Lipofectamine RNAiMax (Invitrogen, ThermoFisher) following the procedure recommended by the manufacturer. The sequences of the siRNA oligonucleotides used were GUGCGAGGGGGUUGUAAUCTT (siCtrl), GCUGCAGCAGAAGAGGCCATT (siEXOSC10) and GGAAUACCAGCUUU CACUUT (siDIS3). Knockdown efficiency was analysed by RT-qPCR 48 h and 72 h after transfection. To this end, total RNA was extracted with TRIzol reagent (Ambion, ThermoFisher), retrotranscribed with SuperScript III (Invitrogen, ThermoFisher) and analysed by qPCR using KAPA SYBR Fast qPCR Master Mix (Kapa Biosystems) in a QIAGEN Rotor-Gene Q. Western blotting was performed to analyse the effect of the siRNA treatments on protein levels.

**Clonogenic assay**. HeLa cells were seeded in six-well plates and transfected with siRNAs to knock down exosome subunits. 3 days after the siRNA transfection, $3 \times 10^5$ cells per condition were seeded and irradiated with 0, 2, 4, or 6 Gy. The cells were grown for 1 week and stained with crystal violet for 30 min. The excess of crystal violet was removed by washing the plates with water and the cells were finally air-dried. Colonies were counted with Cell Profiler[TM 52].

**Antibodies**. The primary antibodies used for western blotting and immunofluorescence were: mouse anti-EXOSC10 (sc-374595, Santa Cruz Biotechnology, dilution 1:100 for IF and 1:1000 for WB), rabbit anti-EXOSC10 (ab50558, dilution 1:500), rabbit anti-DIS3 (ab176802, Abcam, dilution 1:100), mouse anti-Tubulin (Sigma, T5168, dilution 1:50), rabbit anti-γH2AX (#9718, Cell Signaling Technology, dilution 1:400), mouse anti-γH2AX (ab26350, Abcam, dilution 1:500), rabbit anti-RAD51 (ab63801, Abcam, dilution 1:100), mouse anti-RPA (MABE285, Merck Millipore, dilution 1:500), mouse anti-CtIP (61141, Active Motif, dilution 1:400), rat anti-BrdU (B8434, Sigma-Aldrich, dilution 1:250), and mouse monoclonal anti-BrdU (RPN20AB, Sigma, dilution 1:100). Fluorophore-conjugated secondary antibodies used for immunofluorescence were: goat anti-mouse Alexa594 (115-585-003, Jackson Immunoresearch, dilution 1:100), donkey

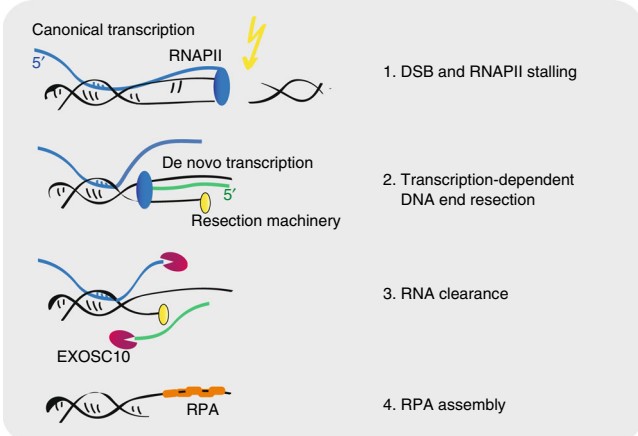

**Fig. 8** Transcription and RNA degradation at DSBs. The figure presents a model for the role of transcription and RNA degradation at DSBs. See the main text for details

anti-rabbit-FITC (711-096-152, Jackson Immunoresearch, dilution 1:100), goat anti-rat Alexa647 (112-605-143, Jackson Immunoresearch, dilution 1:100). Secondary antibodies for western blotting were: HRP-conjugated goat anti-mouse (P0447, Dako, dilution 1:1000) and HRP-conjugated goat anti-rabbit (P0448, Dako, dilution 1:2000).

**Irradiation**. HeLa cells were either exposed to ionizing radiation with a $^{137}$Cs Gammacell 1000 source (Nordion) at the specified doses or micro-irradiated using a wide field Angström's microscope (Leica) equipped with a Micropoint pulsed dye laser of 365 nm (Photonic instruments, Inc.). For micro-irradiation experiments, the cells were seeded in round 25 mm coverslips and cultured overnight in the presence of 10 μM BrdU before laser micro-irradiation.

**Immunofluorescence**. Cells were fixed for 10 min with 3.6% formaldehyde, permeabilized with 0.1% Triton X-100 for 15 min, and blocked for 30 min with 5% bovine serum albumin (BSA) in phosphate buffered saline (PBS). Antibodies were diluted in 1% BSA in PBST (PBS containing 0.01% Tween-20) and antibody incubations were for 1 h. Coverslips were mounted using Vectashield mounting medium with DAPI (VectorLabs) and the slides were visualized in a LSM780 confocal microscope (Carl Zeiss) with an optical thickness of 0.9 μm. Quantitative analyses of the number of cells with foci or stripes were carried out in random areas using FIJI[53]. The number of stripes was quantified in 20–30 cells per preparation. Statistical significance was assessed using the Mann–Whitney's t-test. When stated, cells were pre-permeabilized with CSK buffer (10 mM PIPES, 300 mM sucrose, 100 mM NaCl, 3 mM MgCl$_2$, 1 mM EGTA) for 10 min and treated with RNase A (0.1 mg/ml) for 30 min at 37 °C before fixation. For BrdU immunofluorescence, cells were washed with CSK buffer for 10 min before fixation.

**Flow cytometry**. U2OS cells that were stably transfected with the DR-GFP and EJ5-GPF DNA repair reporters[34,35] were transfected with siRNA as described above. After 48 h, the cells were transfected again with a I-SceI expression plasmid using Lipofectamine 3000 to induce the expression of the I-SceI endonuclease that cleaves the reporter constructs. A second dose of siRNA was given to the cells in the same transfection mixture. The cells were incubated for additional 92 h. The cells were then harvested in PBS, and fixed in suspension with 70% ethanol for 2 h while rotating at 4 °C. The fixed cells were resuspended in 500 μl PBS containing 12.5 μl RNase A (stock solution 10 mg/ml) and 5 μl of propidium iodide (1 mg/ml), and incubated for 30 min at 37 °C. The percentage of GFP-positive cells was analysed by flow cytometry in a FACSCalibur (BD Biosciences) using the FlowJo software (BD Biosciences). For cell cycle analysis, the cells were fixed with 70% ethanol at 4 °C for 2 h, treated with RNase A, stained with propidium iodide and analysed in a FACSCalibur (BD Biosciences).

**Western blotting**. Protein levels were determined by western blotting following standard procedures. Cells were lysed in 2x Laemmli sample buffer and resolved by sodium dodecyl sulfate polyacrylamide gel electrophoresis. Proteins were transferred to PVDF membranes using a semi-dry transfer procedure. Secondary antibodies were HRP conjugates. Blot imaging and quantification were done in a Biorad ChemiDoc XRS + system using the Image Lab 6.0.0 Software (BioRad) for band identification and quantification. Tubulin was used as a loading reference.

**Resection measurements**. DIvA cells were treated for 4 h with 300 nM hydroxytamoxifen (4-OHT, Sigma) to induce the translocation of the AsiSI endonuclease into the cell nucleus. Genomic DNA was isolated with phenol:chloroform and digested with 1 U of Fast Digest enzymes (ThermoFisher) BsrGI and HindIII[37]. Mock-digested samples were processed in parallel. The percentage of ssDNA adjacent to the DSB was measured by quantitative PCR (qPCR) using KAPA SYBR Fast qPCR Master mix (Kapa Biosystems) with the primers indicated in the Supplementary Table 1 in a QIAGEN Rotor-Gene Q. The following equation was used to calculate the percentage of ssDNA:

$$\%ssDNA = \frac{1}{2^{\Delta Ct-1} + 0.5} * 100$$

For each sample, the ΔCt was calculated by subtracting the Ct value of the mock-digested sample from the Ct of the digested sample.

**Single molecule analysis of resection tracks (SMART)**. U2OS cells were grown for 24 h in the presence of 10 μM BrdU, irradiated with ionizing radiation at 10 Gy and incubated for 1 h. Genomic DNA was extracted by gently embedding cells in agarose plugs and resuspended in 1 x MES Buffer (Sigma), pH 5.7. DNA fibres were stretched using a FiberComb Molecular Combing System (Genomic Vision) onto silanized coverslips (Genomic Vision) and dried for 2 h at 65 °C. The stretched fibres were immunostained using an anti-BrdU antibody to visualize the ssDNA. Images were captured with a Nikon Eclipse Ni-E microscope (Nikon) with automatized stage and a 40x objective and processed with IS ELEMENTS Nikon software. For each experiment, at least 200 ssDNA per sample were measured with Adobe Photoshop CS4.

**Strand-specific RT-qPCR (ssRT-PCR)**. Site-specific DSBs were produced in HeLa cells by transfecting the cells with the pOPRSVI/MCS-I-PpoI plasmid, which codes for the I-PpoI endonuclease, with Lipofectamine 3000 transfection kit (Invitrogen, Thermofisher)[25]. The cells were incubated for 20 h to allow for I-PpoI expression. Site-specific DSBs in DIvA cells were produced by incubating the cells with 300 nM 4-OHT for 4 h. Total RNA was extracted with TRIzol reagent (Ambion, ThermoFisher). A total of 0.5 μg of RNA was retrotranscribed with SuperScript III (Invitrogen) using strand-specific primers designed to retrotranscribe RNAs produced upstream and downstream of the DSBs. A primer for ARPP was included in all reverse-transcription reactions together with the DSB-specific primers for normalization purposes. The resulting cDNAs were used for qPCR using KAPA SYBR Fast qPCR Master Mix (Kapa Biosystems) in a QIAGEN Rotor-Gene Q and normalized to ARPP. The primers, listed in Supplementary Table 1, were manufactured by Life Technologies Europe BV. Amplification efficiencies and melting curves obtained with each primer pair were used for quality assessment. The results presented are compiled data from at least three independent biological replicates, each analysed in duplicate. For each experiment, the number of independent replicates is provided in the figure legend.

**DNA–RNA hybrid immunoprecipitation (DRIP)-qPCR**. Cells were seeded in 60-mm plates, transfected with siRNA and with the pOPRSVI/MCS-I-PpoI plasmid to induce site-specific DSBs. After 20 h, the cells were resuspended with lysis buffer (1x TE, 0.05% SDS and 0.05 mg/ml Proteinase K) and incubated at 37 °C overnight. Total genomic DNA, including DNA–RNA hybrids, was extracted using phenol: chloroform:isoamylalcohol (PCI) following standard procedures. A total of 5 μg of genomic DNA were digested with HindIII, EcoRI, BsrGI, XbaI, and SspI restriction enzymes overnight at 37 °C. Negative control samples were also digested with 10 U RNase H1. Digested DNA was diluted in 500 μl binding buffer (10 mM NaPO$_4$, 140 mM NaCl and 0.05% Triton X-100). A total of 7.5 μg of S9.6 antibody (MABE 1095, Merck) were added to the DNA solution and incubated under rotation at 4 °C during 4 h. Immunoprecipitation of the DNA–RNA–antibody complex was performed with 30 μl of Dynabeads Protein A (ThermoFisher) for 2 h at 4 °C with rotation. The beads were washed with binding buffer three times for 10 min each, and the bound material was eluted for 10 min in 50 μl of elution buffer (50 mM Tris-HCl pH 8, 10 mM EDTA and 0.5% SDS). A total of 2.5 μl of Proteinase K (1 mg/ml) were added to the eluted fraction and incubated for 45 min at 55 °C. DNA–RNA hybrids were finally extracted following a standard phenol–chloroform–isoamylalcohol protocol and analysed by qPCR with RYR2 up, RYR2 dw, 28S up, 28S down and GADPH primers. The same sequences were quantified in parallel in the input samples, and DRIP levels were expressed as percentage of input in each sample. The percentage of input was normalized to GADPH signal.

**Small RNA-seq library preparation**. siRNA transfections in HeLa cells were carried out as described above and 48 h after siRNA treatment, I-PpoI was transfected using Lipofectamine 3000 as suggested by the supplier. Total RNA was extracted using TRIzol 36 h after I-PpoI transfection. RNA samples were stored at −80 °C and samples from three independent experiments were processed in parallel for next-generation sequencing. The libraries were prepared using the Illumina TruSeq small RNA library preparation protocol (version December 2014). Small RNA spike-ins (Exiqon, product no. 800100) were dissolved in 150 μl and 1 μl of the spike-in solution was added to 1 μg of total RNA input before library preparation. The barcoded cDNA libraries were pooled and run on an Illumina NextSeq500 sequencer for 75 cycles.

**Analysis of small RNA expression in the 28S rDNA locus**. Analysis was performed using the curated rRNA library from Bonath et al.[10] Adapter sequences were removed and high-quality reads with length longer than 18 nt were processed further. Identical reads were collapsed keeping the read count in the identifier. Then reads were mapped to the custom rDNA library using Bowtie v1.12 [http://bowtie-bio.sourceforge.net] with options '-v 0 -q -k 1 –best". Overall read counts were normalized to spike-ins. The following analyses were performed in R v3.2.2 [www.r-project.org]. For coverage plots, the distance to the cut site was calculated for each read. Subsequently, the nucleotide density at each position upstream or downstream of the cut site was determined and normalized to spike-ins. The average of the three replicates was plotted in 20 nt bins in respect to the I-PpoI site. Read length analysis was performed on collapsed reads mapping upstream antisense to the I-PpoI site. Statistical testing of read counts in siEXOSC10 and siDIS3 samples compared to control siRNA was carried out using a paired two-tailed Student's t-test.

**Cloning and site-specific mutagenesis**. The EXOSC10 cDNA was amplified by PCR using PfuUltra II Fusion HS DNA polymerase (Agilent Technologies) and cloned into the pOPRSVI/MCS vector using ClaI and NotI restriction sites. Amino acid mutations were done directly to the pOPRSVI/MCS-EXOSC10 plasmid by oligonucleotide-directed site-specific mutagenesis using the Site-Directed Mutagenesis kit (Invitrogen). The oligonucleotides used to generate EXOSC10-D313A-E315A mutation were 5′-CCTGTAAGAGTGGTGCGCCAAGGCAACTGCAAAT

TCCTG-3′ and 5′-CAGGAATTTGCAGTTGCCTTGGCGCACCACTCTTACA
GG-3′. The resulting mutant protein was named EXOSC10$^{dea}$.

**Transfection of HeLa cells**. HeLa cells were seeded at 70–80% confluence in DMEM supplemented with 10% FBS without antibiotics and transfected using with Lipofectamine 3000 (Invitrogen) as described in the manufacturer's protocol. The plasmids used were pOPRSVI/MCS-EXOSC10$^{wt}$ and pOPRSVI/MCS-EXOSC10$^{dea}$ for expression of recombinant EXOSC10 proteins, pcDNA3-RNaseH1for RNase H1 expression[54] and pcDNA3 (Invitrogen) as a control "empty" plasmid. The cells were harvested 24 h after transfection unless otherwise stated.

**Chromatin immunoprecipitation (ChIP-qPCR)**. HeLa cells depleted of either DIS3 or TFIIS were transfected with the pOPRSVI/MCS-I-PpoI plasmid and harvested 20 h after transfection. The cells were fixed at room temperature for 10 min by the addition of a fixing solution containing formaldehyde to a final concentration of 2%. Chromatin was extracted, sonicated in a Bioruptor (Diagenode) and immunoprecipitated with anti-RNAPII-CTD antibody (mouse ab5408, abcam) following standard procedures. The immunoprecipitated DNA was purified using Zymo-ChIP Clean & Concentrator (Zymo Research) and analysed by qPCR using the following primers: *RYR2* up, *RYR2* dw, *28S* up, *28S* dw and ARPP. The results were expressed as percentage of input and normalized to ARPP. The results presented are compiled data from three independent biological replicates.

**Statistical analysis**. Histograms show average values and error bars represent standard errors of the mean (s.e.m.). The number of biological replicates for each experiment are indicated in the figure legends. One-sample or two-sample *t*-tests were used for statistical testing of results from ssRT-qPCR, DRIP-qPCR, and resection analyses. The one-sample test was used to establish whether fold changes were significantly different from 1. The Mann–Whitney's test was used for statistical testing of differences in immunofluorescence and SMART experiments. The tests used in each case and the number of replicates are indicated in the figure legends.

**PCR primer sequences**. The sequences of all the primers used in this study are provided in the Supplementary Table 1.

**Reporting summary**. Further information on research design is available in the Nature Research Reporting Summary linked to this article.

## Data availability
The small RNA-seq data are available in the NCBI Gene Expression Omnibus (GEO) under the accession number GSE113109. The source data underlaying Figs. 1b–f, 2a–c, 3a–c, 4b–d, 5a–c, 6a–f, 7a–c, and Supplementary Figs. 1–10 are provided as a Source Data file. All data is available from the authors upon reasonable request.

## Code availability
Custom scripts used for the small RNA-seq analysis are available from the authors upon request.

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

## Acknowledgements

We thank Gaelle Legube for making DIvA cells available to us, Andres Aguilera for the RNase H1 overexpression plasmid, and Inna Biryukova and Anaya Mukherjee for technical assistance. This work was supported by grants from The Swedish Research Council (grants 2015-04553 to N.V. and 1403003 to M.F.), The Swedish Cancer Society (grant CAN 2016/460 to N.V.), and the Spanish Ministry of Economy and Competitivity (R + D + I project grant SAF2016-74855-P to P.H.). J.D.P. and M.E. were supported by the Department of Molecular Biosciences, the Wenner-Gren Institute at Stockholm University. J.D.P. and R.P.C. were recipients of short-term EMBO fellowships (STF-7513-2018 and STF-7764-2018). F.B. was supported by SFO funding from the Faculty of Science at the Stockholm University.

## Author contributions

J.D.P., M.E.C., and N.V. designed the study and planned the experiments. J.D.P., M.E.C., F.B., S.J., and R.P.C. carried out experimental work and interpreted the data. M.R.F. and P.H. supervised the small RNA-seq and SMART experiments, respectively. J.D.P. and N.V. drafted the manuscript. All authors discussed the results and contributed to the final manuscript.

## Additional information

**Competing interests:** The authors declare no competing interests.

