## [Peer Review File · Nature Communications]

Reviewers' comments:

Reviewer #1 (Remarks to the Author):

The authors have done a good job to answer and clarify the issues. Their data are solid and make sense. The manuscript is informative and will be of interest to the community.

I have one suggestion for the abstract. The authors write "DNA end resection is hyperstimulated in EXOSC10-depleted cells." Then, in the next sentence: "The DNA end resection defect". Defect usually refers to abrogation/decrease etc., maybe "misregulated" would be more appropriate, as used later in the text.

Reviewer #2 (Remarks to the Author):

The authors have responded adequately to this reviewer's concerns.

Reviewer #3 (Remarks to the Author):

The paper by Domingo-Prim et al. proposes the transcription-dependent recruitment of the catalytic subunits of the exosome, EXOSC10 and DIS3, to sites of DNA damage, where they modulate dilncRNA levels and facilitate repair by homologous recombination.

Since many of the criticisms raised have not been fully addressed, I think this paper still needs to be improved before publication. Also in this version, only the number of cells analysed and not the number of biological replicates from where the n of cells comes from has been indicated (see 1b, 1c, 2c, 3a, 6b, 6c, 6d, 7c); in Fig. 2b n is not indicated. The same applies for many of the supplementary figures.

Important issues are the following:

- according to the model, transcription is required for DNA end resection and RPA loading but accumulation of these transcripts in EXOSC10 KD cells impairs RPA loading. It is possible that these events, apparently contradictory, are timely regulated, but this is not demonstrated. A time course analysis of the DNA end resection markers, in particular RPA and BrdU stainings, upon EXOSC10 KD and transcriptional inhibition is necessary to support this model. Moreover, the observation that transcription is required for DNA end resection and RPA loading is a strong claim compared to the modest differences shown, this should be toned down.
- In Supplementary Fig. 8 the authors show that catalytically inactive EXOSC10 is recruited to laser stripes. How is it possible that the nuclear staining of EXOSC10 is unchanged in siCTRL and siEXOSC10 (panel 1 and 2) conditions? Since the knock-down is expected to be working, as shown in supplementary Fig. 1, what is the antibody detecting? Moreover, it is not clear why the EXOSC10 staining is so different between Fig. 1a and b. In the first a nucleolar staining is detected, but this is absent in the second, as well as in Supp. Fig. 1. All thus points to the possibility of an unspecific signal.
- To claim that the catalytic activity of EXOSC10 is necessary for DNA repair by HR (page 5 line 162) it is important to show that overexpression of the wt EXOSC10 and not the catalytically inactive one in EXOSC10 KD cells rescues not only RPA loading but also RAD51 loading and,

eventually, repair by HR.

More specific points are the following:

Figure 1D,E: As already mentioned, 48+92h is a timing that could lead to indirect and unspecific results. The authors claim this is in line with published literature (ref 34, 35) but I could not really find those timings in the papers referred. Many other papers use shorter timings for this experiment and I am convinced that also the author could see enough GFP signals at earlier times.

Additionally, the reader cannot really appreciate how much GFP is accumulating since data are normalized on the control and not expressed as %.

Moreover, line 155 page 5 reads that cell cycle analyses have been performed to exclude cell cycle variations. Are the timing the same as in Fig. 1D,E?

Figure 4: The authors claim that EXOSC10 controls diIncRNA levels, although there is no direct evidence for this. Is EXOSC10 recruited to the sites analysed? Is the increase of diIncRNAs rescued by overexpression of wt and not by the catalytically inactive EXOSC10?

Supp. Fig. 2B. Figure legend reads: "The percentage of cells that showed RPA-positive foci were quantified" while on the y-axes: "RPA foci per cell"; the numbers seem low to reflect either the first or the second statement. To have an idea of which kind of foci have been quantified here, images should be shown. (N=45 from how many experiments? This should be always stated). The same experiment should be shown in parallel for cells treated with RNA Pol II inhibitors and images need to be shown.

Figure 6B: The rescue of RPA in EXOSC10 KD cells with RNase A and RNase H1 is very interesting, but BrdU staining should be shown as a control.

Poin-by-point reply to the reviewers' comments

Reviewer #1

I have one suggestion for the abstract. The authors write "DNA end resection is hyperstimulated in EXOSC10-depleted cells." Then, in the next sentence: "The DNA end resection defect". Defect usually refers to abrogation/decrease etc., maybe "misregulated" would be more appropriate, as used later in the text.

REPLY: We agree. The sentence has been modified as suggested.

Reviewer #3

Also in this version, only the number of cells analysed and not the number of biological replicates from where the n of cells comes from has been indicated (see 1b, 1c, 2c, 3a, 6b, 6c, 6d, 7c); in Fig. 2b n is not indicated. The same applies for many of the supplementary figures.

REPLY: The number of cells and number of experiments have been indicated in all figure legends.

Important issues are the following:

- according to the model, transcription is required for DNA end resection and RPA loading but accumulation of these transcripts in EXOSC10 KD cells impairs RPA loading. It is possible that these events, apparently contradictory, are timely regulated, but this is not demonstrated. A time course analysis of the DNA end resection markers, in particular RPA and BrdU stainings, upon EXOSC10 KD and transcriptional inhibition is necessary to support this model.

REPLY: We agree that a time course analysis of the series of events that follow the generation of a DSB would provide very valuable knowledge. However, we expect that the relevant time frames would be very short (based on the fact that DDR foci assembly is very rapid) and, unfortunately, we lack a suitable experimental system for kinetic analyses of this type. Neither the I-PpoI transfection setup nor the I-SceI in DiVA cells allow for precise and synchronized time control.

Moreover, the observation that transcription is required for DNA end resection and RPA loading is a strong claim compared to the modest differences shown, this should be toned down.

REPLY: We agree that there is some resection left and we have rephrased the text to claim that "transcription facilitates DNA end resection" (lines 360 and 374).

- In Supplementary Fig. 8 the authors show that catalytically inactive EXOSC10 is recruited to laser stripes. How is it possible that the nuclear staining of EXOSC10 is unchanged in siCTRL and siEXOSC10 (panel 1 and 2) conditions? Since the knock-down is expected to be working, as shown in supplementary Fig. 1, what is the antibody detecting?

REPLY: Thanks for pointing this out. The knock down is not 100% efficient and there is staining left in approx. 20% of the cells, as shown in the quantification presented in the Supplementary Figure 8. These were not the most representative images. The images have been replaced to better reflect the result of the quantitative analysis.

Moreover, it is not clear why the EXOSC10 staining is so different between Fig. 1a and b. In the first a nucleolar staining is detected, but this is absent in the second, as well as in Supp. Fig. 1. All this points to the possibility of an unspecific signal.

REPLY: In untreated cells, EXOSC10 is heavily concentrated in the nucleolus, as shown in Figure 1A (-Act). The nucleolar localization is likely due to the function of the exosome in pre-rRNA processing. This pattern of staining is specific and is in agreement with data from many others. In some experiments, we have treated the cells with a low dose of ActD (known to inhibit RNAPI but not RNAPII) to inhibit pre-rRNA synthesis. In these conditions, EXOSC10 is no longer accumulated in the nucleolus which facilitates the imaging of microirradiated stripes. The figure legends have been revised to indicate if the cells were treated with low concentration of ActD or not.

- To claim that the catalytic activity of EXOSC10 is necessary for DNA repair by HR (page 5 line 162) it is important to show that overexpression of the wt EXOSC10 and not the catalytically inactive one in EXOSC10 KD cells rescues not only RPA loading but also RAD51 loading and, eventually, repair by HR.

REPLY: This is not what we claim in the sentence of page 5. The sentence tells that EXOSC10 is the exosome subunit that is necessary for DSB repair by HR. We realize that the phrasing was confusing and the text has been rephrased for clarity.

More specific points are the following:

Figure 1D,E: As already mentioned, 48+92h is a timing that could lead to indirect and unspecific results. The authors claim this is in line with published literature (ref 34, 35) but I could not really find those timings in the papers referred. Many other papers use shorter timings for this experiment and I am convinced that also the author could see enough GFP signals at earlier times.

Additionally, the reader cannot really appreciate how much GFP is accumulating since data are normalized on the control and not expressed as %.

REPLY: 48+92 h is a long time but the repair of the induced DSB actually takes place a few hours after I-SceI transfection, i.e. shortly after 48h. The extra time is to allow the accumulation of GFP. The expression of EXOSC10 is no longer relevant in the late phase of the experiment because the repair event has already taken place.

Long incubation times are reported in the literature. Bennardo et al (2008) describe that they added 4OHT 48h after the initiation of siRNA transfection, and that the percentage of GFP+ cells was analyzed by FACS three days after 4OHT was added (see the section “Repair Assays” in the “Materials and Methods” of reference 35, on page 8). Pierce et al. (ref. 34, on page 2637) state that the % GFP positive cells in the population remains stable during expansion of cells for two weeks or longer after DSB induction.

We have replotted the data and replaced the figure to show the percentage of GFP-positive cells in each condition, as required.

Moreover, line 155 page 5 reads that cell cycle analyses have been performed to exclude cell cycle variations. Are the timing the same as in Fig. 1D,E?

REPLY: Yes, the timing of cell cycle analyses were the same as in Figures 1e,f. As explained above, the relevant time point is the time of 4-OHT addition (when the DSBs are produced and repaired), not the time when the GFP signal is analyzed.

Figure 4: The authors claim that EXOSC10 controls dilncRNA levels, although there is no direct evidence for this. Is EXOSC10 recruited to the sites analysed? Is the increase of dilncRNAs rescued by overexpression of wt and not by the catalytically inactive EXOSC10?

REPLY: We have shown the recruitment of EXOSC10 to the analyzed sites by ChIP in a previous publication (Marin-Vicente et al., *J. Cell Sci.* **128**, 1097–1107, 2015). We also show the recruitment of EXOSC10 (wt and mutant) to microirradiated stripes in Fig. 1 and Suppl. Fig. 8.

The reviewer also asks for a rescue experiment for dilncRNAs. We agree that the experiment is interesting and conceptually very valid. However, the rescue experiment proposed by the reviewer would require ssRT-qPCR combined with DSB induction, knock down and overexpression. This is technically very difficult because the levels of dilncRNA expression are extremely low (as we discuss in our recent article Bonath et al., *NAR*, 2018) and small variations have a strong impact on the results. We have carried out this type of experiment and we see the expected trend, but the variability in the data is too large to show statistical significance. Therefore we have chosen not to include the experiment in the manuscript. We present instead other experiments that strongly support our model. We provide evidence that RNA degradation by EXOSC10 facilitates the assembly of RPA at DSBs by showing that the RPA recruitment defect induced by depletion of EXOSC10 is rescued by expression of EXOSC10-wt but not by a catalytically inactive EXOSC10 mutant (Fig. 6b). We also show that dilncRNAs are targets of EXOSC10 (Fig. 4a-c) by analyzing the effect of EXOSC10 depletion on the levels of dilncRNAs at three sequence-specific DSBs (two induced by I-PpoI in HeLa cells and a third one induced by I-SceI in U2OS cells).

Supp. Fig. 2B. Figure legend reads: “The percentage of cells that showed RPA-positive foci were quantified” while on the y-axes: “RPA foci per cell”; the numbers seem low to reflect either the first or the second statement. To have an idea of which kind of foci have been quantified here, images should be shown. (N=45 from how many experiments? This should be always stated).

REPLY: The labeling of the y axis and the description were not correct. The figure has been amended. The revised figure also includes images that illustrate the kind of foci that have been quantified. The number of experiments is stated in the figure legend and the source data are provided as a Source Data file.

The same experiment should be shown in parallel for cells treated with RNA Pol II inhibitors and images need to be shown.

REPLY: A very similar experiment has already been published (see Aymard et al. 2014, doi:10.1038/nsmb.2796) showing that transcription inhibition causes a strong decrease in the recruitment of the homologous recombination machinery to DSBs.

Figure 6B: The rescue of RPA in EXOSC10 KD cells with RNase A and RNase H1 is very interesting, but BrdU staining should be shown as a control.

REPLY: The purpose of these experiments is to show that the RPA recruitment defect observed in EXOSC10-depleted cells is due to the presence of RNA. The fact that two different RNases rescue RPA recruitment strongly supports this point. The reviewer would like us to show that the RNase treatments rescue RPA foci without changing the BrdU signal. However, BrdU would not be a good control in these experiments because the presence of RNA, the assembly of RPA and the extent of DNA resection are related to each other and BrdU levels are thus not expected to be constant. As explained in the Discussion (see lines 432-436 on page 12), RPA plays a complex role in the regulation of DNA end resection and promotes resection termination. Therefore, defects in the assembly of RPA are expected to enhance resection and, in turn, BrdU labeling.

Reviewers' comments:

Reviewer #3 (Remarks to the Author):

- according to the model, transcription is required for DNA end resection and RPA loading but accumulation of these transcripts in EXOSC10 KD cells impairs RPA loading. It is possible that these events, apparently contradictory, are timely regulated, but this is not demonstrated. A time course analysis of the DNA end resection markers, in particular RPA and BrdU stainings, upon EXOSC10 KD and transcriptional inhibition is necessary to support this model.

REPLY: We agree that a time course analysis of the series of events that follow the generation of a DSB would provide very valuable knowledge. However, we expect that the relevant time frames would be very short (based on the fact that DDR foci assembly is very rapid) and, unfortunately, we lack a suitable experimental system for kinetic analyses of this type. Neither the I-Ppol transfection setup nor the I-SceI in DiVA cells allow for precise and synchronized time control.

REPLY: I agree with the author the I-Ppol and DiVA systems may not be suitable for this experiments; however a very viable and valuable alternative is immunostaining of the DNA end resection markers (RPA and BrdU stainings) upon EXOSC10 KD and transcriptional inhibition at different time-points after ionizing radiations.

- Supp. Fig. 2B. Figure legend reads: "The percentage of cells that showed RPA-positive foci were quantified" while on the y-axes: "RPA foci per cell"; the numbers seem low to reflect either the first or the second statement. To have an idea of which kind of foci have been quantified here, images should be shown. (N=45 from how many experiments? This should be always stated).

REPLY: The labeling of the y axis and the description were not correct. The figure has been amended. The revised figure also includes images that illustrate the kind of foci that have been quantified. The number of experiments is stated in the figure legend and the source data are provided as a Source Data file.

REPLY: The time-point from which the images shown come from is not specified and apparently it was not clear that images from all the time-points should have been shown. Nonetheless, it is quite confusing that at maximum 5 RPA foci colocalize with H2AX.

The same experiment should be shown in parallel for cells treated with RNA Pol II inhibitors and images need to be shown.

REPLY: A very similar experiment has already been published (see Aymard et al. 2014, doi:10.1038/nsmb.2796) showing that transcription inhibition causes a strong decrease in the recruitment of the homologous recombination machinery to DSBs.

I am aware of the Aymard et al. experiment. However, this experiment exploits the AsiSi system, which, as the author pointed out above, is not a good system for studying dynamics of events at DSBs. Showing the dynamics of events at DSBs as suggested by immunostaining of the DNA end resection markers at IR-foci remains a valuable experiment and surely would strengthen this manuscript. It is surprising that the authors seem unwilling to generate these data.

On a general note. Dots overlapping the bar graphs have not been added to all the figures and the author should not forget to add those to the supplementary figures as well.

1) according to the model, transcription is required for DNA end resection and RPA loading but accumulation of these transcripts in EXOSC10 KD cells impairs RPA loading. It is possible that these events, apparently contradictory, are timely regulated, but this is not demonstrated. A time course analysis of the DNA end resection markers, in particular RPA and BrdU stainings, upon EXOSC10 KD and transcriptional inhibition is necessary to support this model.

REPLY: We agree that a time course analysis of the series of events that follow the generation of a DSB would provide very valuable knowledge. However, we expect that the relevant time frames would be very short (based on the fact that DDR foci assembly is very rapid) and, unfortunately, we lack a suitable experimental system for kinetic analyses of this type. Neither the I-PpoI transfection setup nor the I-SceI in DivA cells allow for precise and synchronized time control.

REVIEWER'S REPLY: I agree with the author the I-PpoI and DivA systems may not be suitable for this experiments; however a very viable and valuable alternative is immunostaining of the DNA end resection markers (RPA and BrdU stainings) upon EXOSC10 KD and transcriptional inhibition at different time-points after ionizing radiations.

REPLY:

The reviewer requires that we look into the dynamics of events that take place at DNA double strand breaks. More specifically, the experiment would imply immunostaining of DNA end resection markers (RPA and BrdU stainings) upon EXOSC10 KD and transcriptional inhibition at different time-points after ionizing radiation. The coordination in time of the different events that take place at DSBs is a very important question, but we do not have suitable tools to address it and we do not think that the experiment proposed by the reviewer will resolve it. It is true that ionizing radiation is a relatively fast procedure compared to other means of producing DNA damage, but it takes 3-5 min to irradiate the plates (depending on the doses that we select), which means that different DSBs are not synchronized in the cell population. This lack of synchronization within a 3-5 min time-frame is problematic because the DNA repair machinery assembles within 5 min (see for example Polo & Jackson 2011, doi: [10.1101/gad.2021311](https://doi.org/10.1101/gad.2021311)). Another important problem with the proposed setup is that by inhibiting transcription, we not only inhibit the synthesis of the dilncRNAs, but also inhibit DNA end resection (as shown in the manuscript, Figure 7), which in turn impairs RPA assembly regardless of the activity of EXOSC10. Therefore, the use of transcription inhibitors will not support any conclusions about the role of EXOSC10 at DSBs. In summary, we don't think this experiment will give any meaningful outcome.

2) Supp. Fig. 2B. Figure legend reads: "The percentage of cells that showed RPA-positive foci were quantified" while on the y-axis: "RPA foci per cell"; the numbers seem low to reflect either the first or the second statement. To have an idea of which kind of foci have been quantified here, images should be shown. (N=45 from how many experiments? This should be always stated).

REPLY: The labeling of the y axis and the description were not correct. The figure has been amended. The revised figure also includes images that illustrate the kind of foci that have been quantified. The number of experiments is stated in the figure legend and the source data are provided as a Source Data file.

REVIEWER'S REPLY: The time-point from which the images shown come from is not specified and apparently it was not clear that images from all the time-points should have been shown. Nonetheless, it is quite confusing that at maximum 5 RPA foci colocalize with H2AX.

REPLY:

- The figure has been revised and images from all times are shown.
- The time points from which the images come from are indicated.
- The low number of foci is due to the high cutoff used in these experiments (to avoid counting RPA located at sites different from DDR foci, for example replication forks). The important point is not the absolute number foci, but the relative number in the different conditions. We have re-quantified the data to be sure that the results are correct and we have revised the figure accordingly. The conclusions are the same.

REVIEWER: The same experiment should be shown in parallel for cells treated with RNA Pol II inhibitors and images need to be shown.

REPLY: A very similar experiment has already been published (see Aymard et al. 2014, doi:10.1038/nsmb.2796) showing that transcription inhibition causes a strong decrease in the recruitment of the homologous recombination machinery to DSBs.

REVIEWER'S REPLY: I am aware of the Aymard et al. experiment. However, this experiment exploits the AsiSi system, which, as the author pointed out above, is not a good system for studying dynamics of events at DSBs. Showing the dynamics of events at DSBs as suggested by immunostaining of the DNA end resection markers at IR-foci remains a valuable experiment and surely would strengthen this manuscript. It is surprising that the authors seem unwilling to generate these data.

REPLY:

As explained above and shown in Figure 7, RNA pol II inhibitors inhibit resection and RPA assembly independently of EXOSC10 activity. Therefore the experiment with RNA pol II inhibitors is not meaningful.

On a general note. Dots overlapping the bar graphs have not been added to all the figures and the author should not forget to add those to the supplementary figures as well.

REPLY:

Dots have been added to the supplementary figures.